# Clinical presentation of human leptospirosis in febrile patients: Urabá, Colombia

**Pablo Uribe-Restrepo**[1]*, **Janeth Perez-Garcia**[2], **Margarita Arboleda**[3], **Claudia Munoz-Zanzi**[4], **Piedad Agudelo-Florez**[1]

1 Graduate School, CES University, Medellin, Colombia, 2 Faculty of Veterinary Medicine and Animal Sciences, CES University, Medellin, Colombia, 3 Colombian Institute of Tropical Medicine, Apartadó, Colombia, 4 School of Public Health, University of Minnesota, Minneapolis, Minnesota, United States of America

* uribe.pablo@uces.edu.co

## Abstract

### Background

Leptospirosis is responsible for various clinical syndromes, classically linked with fever and acute kidney injury.

### Methodology/Principal findings

A prospective multicenter observational study was conducted in six health institutions in the region of Urabá, Colombia. Enrollment was based on leptospirosis-compatible clinical syndrome and a positive preliminary serological test, with PCR used to confirm the disease. Clinical data were collected using a standard questionnaire at enrollment, complemented with a review of clinical records. A total of 100 patients were enrolled, 37% (95% CI 27.0–46.9%) had a positive PCR result confirming acute leptospirosis. The most frequent symptoms in patients with a positive PCR test were headache (91.9%; 34/37), chills and sweating (80.6%; 29/37), nausea (75%; 27/37), dizziness (74.3%; 26/37), vomiting (61.1%; 22/37), congestion (56.8%; 21/37), and conjunctival suffusion (51.4%; 19/37). The frequency of clinical signs classically described in leptospirosis was low: jaundice (8.3%; 3/36) and anuria/oliguria (21.6%; 8/37). An increased neutrophile percentage was reported in 60.6% (20/33) of patients. The presence of complications was 21.6% (8/37), with pulmonary complications being the most frequent (75.0% 6/8). One confirmed case died resulting in a fatality of 2.7% (95% CI 0.5–13.8).

### Conclusions/Significance

Leptospirosis should be considered within the differential diagnoses of an undifferentiated acute febrile syndrome. Leptospirosis presents diagnostic challenges due to limitations in both clinical and laboratory diagnosis thus it is important to improve understanding of disease presentation and identify signs and symptoms that might help differentiate it from other causes of febrile illness.

**Data Availability Statement:** All relevant data are within the manuscript and its Supporting Information file.

**Funding:** The original study was conducted with funding from the Colombian Ministry of Science (Minciencias) (project code 122865740423 to PAF). The manuscript was developed with funding from the Dirección de Investigación e Innovación of Universidad CES (Code INV.032022.002 to JPG). The funders had no role in study design, data collection and analysis, decision to publish, or preparation of the manuscript.

**Competing interests:** The authors have declared that no competing interests exist.

## Author summary

Leptospirosis is a Neglected Tropical Disease with worldwide distribution, a heavy impact in tropical regions, and remaining knowledge gaps. Laboratory disease confirmation requires tests that are still difficult to access in rural communities where the disease is most frequent and has the heaviest impact. We conducted a study on febrile patients who approached health services in the region of Urabá in Colombia. Patients presenting with cases compatible with leptospirosis were subjected to a PCR test to confirm the disease. The disease presents with fever, pain, and gastrointestinal symptoms resulting in a highly unspecific febrile syndrome similar to other tropical infections prevalent in the same areas. However, our results suggest that some findings can increase clinical suspicion of leptospirosis and help guide decision-making among medical professionals. Eye pain had a negative correlation with leptospirosis and might guide clinicians toward other diseases, while kidney involvement among children might increase suspicion of leptospirosis in undifferentiated tropical febrile syndrome. It is important to continue studies like this to better define the spectrum of leptospirosis clinical syndromes.

## 1. Introduction

Leptospirosis is a re-emerging zoonotic disease with global distribution, high endemicity in tropical regions, and a wide range of clinical manifestations [1]. It is estimated that leptospirosis affects around 1.03 million people and is responsible for up to 58,900 deaths each year [2]. *Leptospira*-host interaction is responsible for various clinical syndromes, classically linked with fever and acute kidney injury. It has emerged as a cause of respiratory infection and chronic kidney disease [3,4]. Despite being a neglected disease, the World Health Organization does not list it as such [5]. Nonetheless, efforts around the world are being made to advance research and knowledge about the disease [6]. A systematic review of the human leptospirosis seroprevalence in Colombia during 2000–2012 revealed seroprevalence estimates that varied between 6% and 35% depending on the study [7]. Despite this, the Colombian National Institute of Health only reported about 2,500 cases annually over the same period.

In the tropics, febrile syndrome is caused by different infections with overlapping symptoms, and due to diagnostic limitations, a confirmed diagnosis is often not achieved [8]. Clinical presentation of leptospirosis includes two phases, a septicemic phase characterized by unspecific febrile illness followed by an immune phase with generally reversible local organ impairment, where the kidneys and liver are the most frequent targets [9]. Weil's disease is a severe form of leptospirosis associated with fulminant kidney and liver impairment manifested by a triad of jaundice, acute kidney injury (AKI), and hemorrhagic manifestations [10]. *Leptospira* is an emerging culprit in acute respiratory distress syndrome causing dyspnea and hemoptysis; this form of the disease has increased in frequency in recent years [11].

Leptospirosis can be diagnosed by isolation of *Leptospira* through culture, antibody detection through serological tests, or DNA identification by molecular techniques [12]. The Colombian National Institute of Health (INS) defines laboratory-confirmed leptospirosis as a patient with compatible clinical symptoms and any of the following: microagglutination (MAT) seroconversion in paired samples, MAT titers equal or above 1:400 in a single sample for lethal cases, a positive culture, or a positive molecular DNA test such as PCR [13]. Culture may take months to yield positive results and has low sensitivity, reducing its clinical utility. Diagnosing leptospirosis using PCR on a single sample is becoming increasingly necessary,

given the low likelihood of patients returning to the health center to provide a convalescent sample after the resolution of symptoms, particularly among patients with mild illness or rural communities [14,15].

The region of Urabá in Colombia has a reported leptospirosis seroprevalence of 12.5%, yet, there is evidence of low diagnosis and low reporting of cases [16,17]. Improving the clinical understanding of the disease in the region is the first step to developing better clinical guidelines for diagnosis and surveillance. This study was part of a broader eco-epidemiological study on leptospirosis in the Urabá region of Colombia between 2016 and 2017, involving the identification and enrollment of leptospirosis patients at six health centers. It aims to characterize the acute clinical presentation of PCR-confirmed leptospirosis patients.

## 2. Methods

### Ethics statement

The procedures followed in the study were in accordance with international ethical standards following the Helsinki Declaration of 1975, as revised in 1983, and in compliance with the Colombian national regulations in accordance with Resolution 8430 of 1993. The study received approval from the CES University ethics committee for research in humans, Act 70 from March 28, 2014, project code 309. The study used written informed consent by the participant for inclusion, in case of under aged participants a parent had to provide additional written consent.

A prospective multicenter cross-sectional study within six health centers was conducted in the Urabá region of Colombia. Recruitment consisted of identifying all patients seeking medical services between February 2016 and January 2017 at any of the six participating institutions, with a probable case of leptospirosis based on clinical symptoms and a positive preliminary IgM serological test (dual path platform (DPP) or ELISA) [18,19]. A probable case was defined under the Colombian INS as a patient presenting with a fever of fewer than 3 weeks; a suggestive epidemiological history such as exposure to floodings, mud or stagnant water, work exposure (garbage collection, working within water bodies, farm or agricultural work, or contact with sick animals or rodents); and two or more of the following signs or symptoms: headache, myalgia, conjunctivitis, arthralgia, vomiting, diarrhea, back pain, chills, retroocular pain or photophobia, or rash [13]. Informed written consent was required for enrollment, for participants under the age of 18, written consent was required from both patients and a legal guardian. After a patient meeting the inclusion criteria was identified by the participating institution, the research group was notified, and project personnel collected the biospecimens needed for study purposes (whole blood and urine) to complete diagnostic confirmation; therefore, days post onset (DPO) at the time of the initial clinical consultation and DPO at the time of sample collection were not the same. Recruitment continued until enrollment reached 100 patients. PCR was performed on blood and urine samples through the amplification of the 16S ribosomal gene, a fragment of 331 base pairs was amplified using the oligonucleotides F16S (GGCGCGTCTTAAACATGCAAG) and R16S (GAGCAAGATTCT-TAACTGCTGCC) [20].

Clinical and sociodemographic data were obtained from participants using a standardized questionnaire and additional clinical information was extracted from medical records on follow-up. Among social variables, socioeconomic stratum was defined based on the classification of their place of residence. The age limit to differentiate children and adults was set at 19 instead of 18 because that is the average age of graduating school in the region, and exposure factors change the most after this. Data analyses consisted of descriptive summary measures and comparative analyses which were done using Mann Whitney's U test (U-MW) or

Student's t for continuous variables and Fisher test or chi2 test for categorical variables with Yates Correction for small sample size [21]. The prevalence ratio (PR) and 95% confidence interval were calculated to estimate the effect of specific variables on the proportion of cases that were PCR-positive [22]. Statistical significance was set at a p-value < 0.05. All statistical analyses were conducted in SPSS 21.0.0 (IBM, SPSS Tnc., Chicago, Illinois, USA). A final follow-up after hospital discharge was carried out by researchers through a telephone call to document the status of the patient after the acute disease.

## 3. Results

The enrolled leptospirosis probable cases were distributed across the six sites, with 43.0% of patients coming from the largest urban health center and the remaining coming from smaller urban health centers and a reference hospital located on the outskirts of town. Most enrolled patients were male (63.0%, 63/100), adults older than 19 years old (61.0%, 61/100), self-identified as mestizo race (65.0%, 65/100), from the urban area (60.0%, 60/100), and did not report a previous episode of leptospirosis (97.0%, 97/100). All patients reported fever as this was among the inclusion criteria. Based on the self-reported date of illness onset, DPO at the time of the medical visit were ≤7 for 61.0% (61/100) of the patients, 8–14 days for 16.0% (16/100), 15–21 days for 9.0% (9/100), and >21 days for 9.0% (9/100). DPO to medical visit was not established for the remaining 5 patients. Of the 100 enrolled patients, 37 had a positive PCR result, confirming an acute leptospirosis diagnosis proportion of 37% (95% CI: 27.0–46.9%). Of these patients 22 were positive blood samples, 12 were positive urine samples, and three tested positive in both urine and blood samples. Among patients with sample collection for PCR testing during the first week of illness (DPO ≤7 days at the time of sample collection), 31.1% (14/45) had a positive PCR result; 20.0% (9/45) were positive on blood samples and 13.3% (6/45) were positive on a urine sample, including one patient who was positive on blood and urine samples. For those with DPO of 8–14 days at the time of sample collection, 37.5% (9/24) were PCR positive; 20.8% (5/24) were positive on a blood sample and 16.7% (4/24) were positive on a urine sample. For those with DPO of 15–21 days at sample collection time, 18.2% (2/11) were PCR positive, both positive on a blood sample. Among the patients with a sample collection DPO of >21 days, 57.9% (11/19) were PCR positive; 42.1% (8/19) were positive on a blood sample, 26.3% (5/19) were also positive on a urine sample, and 10.5% (2/19) were positive on both samples. One patient without a recorded illness onset date had a positive PCR result on a blood sample.

### 3.1 Factors associated with a PCR-positive result

Patients without formal paid employment (students, children, adolescents, and housewives) exhibited a higher proportion of PCR positivity (PR 1.5, 95% CI: 1.1–2.0) compared to those who reported having a paid occupation. Those who reported indigenous race exhibited a higher proportion of PCR positivity (PR 9.4, 95% CI: 0.3–69.0) compared to those who identified as mestizo, black or white (Table 1). Eye Pain was less frequent among PCR-negative patients (PR 0.46, 95% CI: 0.2–0.9), other symptoms show no statistically significant differences (Table 2). No statistically significant differences among clinical signs (Table 3).

Stratifying by age group showed that 41.0% (16/39) of children and 34.4% (21/61) of adults were PCR positive. Further investigation by age group of factors associated with PCR positivity revealed that children with anuria/oliguria were more than twice as likely to test PCR positive compared with those without (PR 2.33; 95% CI: 1.22–4.45, p = 0.045), but there was no association for adults (p = 1.0). Similarly, the inability to stand was positively associated with PCR positivity in children (PR 2.4; 95% CI: 1.09–5.26, p = 0.023) but not among adults (p = 0.640).

**Table 1. Sociodemographic characteristics of 100 probable leptospirosis patients tested by PCR for diagnosis confirmation, Apartadó, Colombia 2016–2017.**

| Characteristic | PCR Positive | Total | Proportion PCR positive | PCR positivity Prevalence Ratio (95% C.I.) | P-value |
|---|---|---|---|---|---|
| Sex | | | | | |
| Female | 17 | 37 | 45.9% | 1.5 (0.2–2.4) | 0.875 |
| Male | 20 | 63 | 31.7% | 1 | |
| Setting | | | | | |
| Urban | 18 | 60 | 30.0% | 0.6 (0.4–1.1) | 0.076 |
| Rural | 19 | 40 | 47.5% | 1 | |
| Race/ethnicity | | | | | |
| Indigenous | 5 | 9 | 55.5% | 9.4 (0.3–69.0) | 0.024 |
| Mestizo | 28 | 65 | 43.0% | 7.3 (1.1–50.1) | |
| Black | 3 | 9 | 33.3% | 5.7 (0.7–46.9) | |
| White | 1 | 17 | 5.8% | 1 | |
| Age | | | | | |
| Over 20 years | 21 | 61 | 34.4% | 0.8 (0.5–1.4) | 0.503 |
| 19 years or under | 16 | 39 | 41.0% | 1 | |
| Previous episode of leptospirosis | | | | | |
| Yes | 2 | 3 | 66.6% | 1.85 (0.8–4.2) | 0.308 |
| No | 35 | 97 | 36.0% | 1 | |
| Paid employment* | | | | | |
| Yes | 10 | 43 | 23.2% | 1.5 (1.1–2.0) | 0.031 |
| No | 27 | 57 | 47.3% | 1 | |
| Symptomatic person within the household | | | | | |
| Yes | 14 | 36 | 38.8% | 1.1 (0.6–1.8) | 0.894 |
| No | 23 | 64 | 35.9% | 1 | |
| Socio-economic status† | | | | | |
| Low | 32 | 85 | 37.6% | 1.0 (0.5–2.3) | 0.925 |
| Medium | 5 | 14 | 35.7% | 1 | |

NC: Not calculated

*No paid employments included underaged individuals not currently in school, students, housewives, and unemployed people

† Stratum is a socioeconomic classification used in Colombia (derived from the characteristics of the area of residence).

PCR positivity was nearly 3-fold higher in adults with hypoventilation compared to those without (PR 2.75; 95% CI: 1.51–5.02, p = 0.038) but this effect was not found in children (p = 1.0).

Some form of antibiotic treatment was received by 98% (97/99) of the patients, with no statistical differences between PCR-positive and PCR-negative patients. The most common antibiotic treatments were ceftriaxone (58.8%; 57/97) and doxycycline (50.5%; 49/97). Other antibiotics prescribed in smaller proportions included penicillin, ciprofloxacin, ampicillin-sulbactam, and amoxicillin.

### 3.2 Demographic and clinical characteristics of confirmed cases

Among the 37 leptospirosis-confirmed patients, they were frequently male (54.1%; 20/37), over the age of 20 (56.8%; 21/37), and self-recognized as mestizo (75.7%; 28/37). More confirmed cases were from a low socioeconomic stratum (62.2%; 23/37) and rural areas (51.4%; 19/37) (**Table 1**).

Thirty-six patients had their temperature recorded at the time of enrollment with an average age of 37.7˚C, and 55.5% (20/36) of them had a temperature above 37.5˚C. The average heart rate was 95.04 bpm, and the average respiratory rate was 21.35 rpm.

**Table 2. Symptoms self-reported by 100 probable leptospirosis patients tested by PCR for diagnosis confirmation, Apartadó, Colombia 2016–2017.**

| Symptom | Proportion of all enrolled patients with symptom | Proportion PCR positive among patients with the symptom | Proportion PCR positive among patients without the symptom | PCR positivity Prevalence Ratio (95% C.I.) | P-value |
|---|---|---|---|---|---|
| Malaise# | 96/100 (96.0%) | – | – | – | – |
| Headache | 84/100 (84.0%) | 34/84 (40.5%) | 3/16 (18.8%) | – | 0.152* |
| Chills | 78/97 (80.4%) | 29/78 (27.2%) | 7/19 (36.8%) | 1.01 (0.5–1.9) | 0.978 |
| Myalgia | 69/100 (69.0%) | 28/69 (40.6%) | 9/31 (29.0%) | 1.40 (0.7–2.6) | 0.269 |
| Nausea | 68/99 (68.7%) | 27/68 (39.7%) | 9/31 (29.0%) | 1.37 (0.7–2.5) | 0.306 |
| Dizziness | 67/97 (69.1%) | 26/67 (38.8%) | 9/30 (30.0%) | 1.29 (0.6–2.4) | 0.404 |
| Arthralgia | 63/98 (64.3%) | 26/63 (41.3%) | 11/35 (31.4%) | 1.31 (0.7–2.3) | 0.336 |
| Calf Pain | 52/93 (55.9%) | 23/52 (44.2%) | 11/41 (26.8%) | 1.65 (0.9–3.0) | 0.084 |
| Vomit | 56/99 (56.6%) | 22/56 (39.3%) | 14/43 (32.6%) | 1.21 (0.7–2.0) | 0.490 |
| Nasal congestion | 47/99 (47.5%) | 21/47 (44.7%) | 16/52 (30.8%) | 1.45 (0.8–2.4) | 0.153 |
| Inability to stand | 43/98 (43.8%) | 20/43 (46.5%) | 16/55 (29.1%) | 1.60 (0.9–2.7) | 0.076 |
| Low back pain | 50/98 (51.0%) | 19/50 (38.0%) | 17/48 (35.4%) | 1.07 (0.6–1.8) | 0.791 |
| Conjunctival suffusion | 52/100 (52.0%) | 19/52 (36.5%) | 18/48 (37.5%) | 0.97 (0.6–1.6) | 0.912 |
| Cough | 46/97 (47.4%) | 18/46 (39.1%) | 19/51 (37.3%) | 1.05 (0.6–1.7) | 0.849 |
| Sweating | 39/97 (40.2%) | 15/39 (38.5%) | 20/58 (34.5%) | 1.12 (0.6–1.9) | 0.689 |
| Pain in the right hypochondrium | 38/94 (40.4%) | 15/38 (39.5%) | 20/56 (35.7%) | 1.11 (0.6–1.8) | 0.711 |
| Dyspnea | 35/99 (35.4%) | 12/35 (34.3%) | 24/64 (37.5%) | 0.91 (0.5–1.6) | 0.751 |
| Diarrhea | 43/99 (43.4%) | 14/43 (32.6%) | 23/56 (41.1%) | 0.79 (0.5–1.4) | 0.385 |
| Hemorrhagic manifestations | 20/100 (20.0%) | 9/20 (45.0%) | 28/80 (35.0%) | 1.29 (0.7–2.2) | 0.407 |
| Anuria/Oliguria | 15/100 (15.0%) | 8/15 (53.3%) | 29/85 (34.1%) | 1.56 (0.90–2.73) | 0.155 |
| Eye Pain | 37/99 (37.4%) | 8/37 (21.6%) | 29/62 (46.8%) | 0.46 (0.2–0.9) | 0.012 |
| Earache | 17/99 (17.2%) | 7/17 (41.2%) | 30/82 (36.6%) | 1.13 (0.6–2.1) | 0.722 |
| Dysuria | 11/100 (11.0%) | 6/11 (54.6%) | 31/89 (34.8%) | 1.57 (0.8–2.8) | 0.201 |

* Fisher's Exact Test # Prevalence ratio not calculated due to sample size

The most frequent symptoms were headache (91.9%; 34/37), chills and sweating (80.6%; 29/37), nausea (75%; 27/37), dizziness (74.3%; 26/37), vomiting (61.1%; 22/37), congestion (56.8%; 21), and conjunctival suffusion (51.4%; 19). Malaise and musculoskeletal alterations were reported by all confirmed patients. These alterations include myalgia (75.7%; 28/37), arthralgia (70.3%; 26/37), calf pain (67.6%; 23/37), inability to stand (55.6%; 20/37), and low back pain (52.8%; 19/37) (**Table 4**). Other symptoms evaluated are presented in **Table 4**.

The frequency of clinical signs classically described in severe leptospirosis was low; jaundice (8.3%; 3/37) and anuria/oliguria (21.6%; 8/37). Cyanosis and bradycardia were not recorded in any of the febrile patients included in the study. Additional clinical signs are detailed in **Table 4**. There were no significant differences in total complications by age group.

Blood count results were available for 33 patients. Of these, 21.2% (7/33) had elevated ($>$ 11,000 mm3) white blood cell (WBC), and 60.6% (14/33) had a neutrophil count over 60.0%. Platelet count was altered in 30.3% (10/33) patients with 27.3% (9/33) having under 150,000 mm3 platelets and 3.0% (1/33) having over 400,000 mm3 platelets. The mean hemoglobin count was 12.4 g/dL with 48.5% (16/33) patients having hemoglobin levels under 12 g/dL. Another 28 patients had urine test results with 3.6% (1/28) having a urine density under 1005, with a mean urine density of 1014.1. Similarly, 3.6% (1/28) of patients had a urine pH $>$ 8 and no patients had a urine pH $<$5 with a mean urine pH of 5.9. Twenty patients had

**Table 3. Signs identified by the attending physician in 100 leptospirosis probable patients tested by PCR for diagnosis confirmation, Apartadó, Colombia 2016–2017.**

| Sign | Proportion of patients with the sign among all enrolled patients (%) | Proportion PCR positive among patients with sign | Proportion of PCR positive among patients without the sign | PCR positivity PR (95% C. I.) | P-value |
|---|---|---|---|---|---|
| Musculoskeletal system # alterations | 84/84 (100) | – | – | – | – |
| Abdominal tenderness | 46/98 (46.9) | 17/46 (37.0%) | 19/52 (36.5%) | 1.01 (0.6–1.7) | 0.966 |
| Tachycardia | 22/95 (23.2) | 10/22 (45.5%) | 25/73 (34.2%) | 1.33 (0.7–2.3) | 0.339 |
| Pharyngeal congestion | 25/97 (25.8) | 9/25 (36.0%) | 26/72 (36.1%) | 0.99 (0.5–1.8) | 0.922 |
| Rash | 18/98 (18.4) | 8/18 (44.4%) | 28/80 (35.0%) | 1.27 (0.7–2.3) | 0.453 |
| Adenomegaly | 18/97 (18.6) | 8/18 (44.4%) | 28/79 (35.4%) | 1.25 (0.6–2.2) | 0.476 |
| Hepatomegaly | 18/97 (18.6) | 7/18 (38.9%) | 28/79 (35.4%) | 1.10 (0.5–2.1) | 0.784 |
| Conjunctival alterations | 15/99 (15.2) | 7/15 (46.7%) | 29/84 (34.5%) | 1.35 (0.7–2.5) | 0.368 |
| Splenomegaly | 6/98 (6.1) | 4/6 (66.7%) | 31/92 (33.7%) | 1.98 (1.0–3.7) | 0.118* |
| Meningeal irritation | 7/99 (7.1) | 4/7 (57.1%) | 32/92 (34.8%) | 1.64 (0.8–3.3) | 0.215* |
| Nasal disturbances | 8/96 (8.3) | 3/8 (37.5%) | 31/88 (35.2%) | 1.07 (0.4–2.7) | 0.588* |
| Rhonchi | 5/98 (5.1) | 3/5 (60.0%) | 32/93 (34.4%) | 1.74 (0.8–3.7) | 0.242* |
| Jaundice | 11/97 (11.3) | 3/11 (27.3%) | 33/86 (38.4%) | 0.71 (0.2–1.9) | 0.358* |
| Rales | 9/98 (9.2) | 3/9 (33.3%) | 33/89 (37.1%) | 0.90 (0.3–2.3) | 0.566* |
| Oral cavity lesions# | 4/97 (4.1) | – | – | – | – |
| Wheezing | 6/99 (6.1) | 2/6 (33.3%) | 34/93 (36.6%) | 0.91 (0.2–2.9) | 0.691* |
| Seizures# | 3/99 (3.0) | – | – | – | – |
| Petechiae# | 1/92 (1.1) | – | – | – | – |

* Fisher's Exact Test

# Prevalence ratio not calculated due to sample size.

a recorded creatinine test result and 15.0% (3/20) had a creatinine level above 1.1 mg/dL. A total of 13 patients had a reported total bilirubin (TB) score with 38.5% (5/13) having levels over 1.2 mg/dL and 12 patients having direct bilirubin scores (DB), with 100% (12/12) having a relation DT/TB above 30.0%. CPK was elevated in 35.0% of patients (7/20). Additional laboratory findings are presented in **Table 5**.

Antibiotic treatment was prescribed for 97.8% (35/37) of the patients with an average treatment duration was 11 days (SD 12.3 days). 5.7% (2/35) did not complete it as prescribed by the attending physician. One of these patients died and the reason for not completing treatment for the second patient was not established.

Hospital stay was necessary for 83.8% (31/37) of the leptospirosis patients, with an average stay of 7.0 days (SD 4.1) and 12.9% (4/31) of these patients requiring admission to an Intensive

**Table 4. Signs and Symptoms in leptospirosis patients confirmed by PCR, Apartadó, Colombia 2016–2017.**

| Sign/Symptom | Positive | Reported | Frequency | Sign/Symptom | Positive | Reported | Frequency |
|---|---|---|---|---|---|---|---|
| **Flu-like Symptoms** | | | | **Pulmonary Syndrome** | | | |
| Malaise | 37 | 37 | 100% | Dyspnea | 12 | 36 | 33.3% |
| Chills | 29 | 36 | 80.6% | Wheezing | 2 | 36 | 5.6% |
| Myalgia | 28 | 37 | 75.7% | Rhonchi | 3 | 35 | 8.6% |
| Nausea | 27 | 36 | 75.0% | Rales | 3 | 36 | 8.3% |
| Dizziness | 26 | 35 | 74.3% | **Other** | | | |
| Arthralgia | 26 | 37 | 70.3% | Musculoskeletal system alterations | 32 | 32 | 100.0% |
| Vomit | 22 | 36 | 61.1% | Abdominal tenderness | 17 | 36 | 47.2% |
| Nasal congestion | 21 | 37 | 56.8% | Splenomegaly | 4 | 35 | 11.4% |
| Cough | 18 | 37 | 48.6% | Nasal disturbances | 3 | 34 | 8.8% |
| Tachycardia | 10 | 35 | 28.6% | Oral cavity lesions | 2 | 35 | 5.7% |
| Pharyngeal congestion | 9 | 35 | 25.7% | Low back pain | 19 | 36 | 52.8% |
| Rash | 8 | 36 | 22.2% | Sweating | 15 | 35 | 42.9% |
| **Weil's disease** | | | | Right hypochondrium pain | 15 | 35 | 42.9% |
| Hemorrhagic manifestations | 9 | 37 | 24.3% | | | | |
| Dysuria | 6 | 37 | 16.2% | Earache | 7 | 37 | 18.9% |
| Jaundice | 3 | 36 | 8.3% | Adenomegaly | 8 | 36 | 22.2% |
| Petechiae | 1 | 34 | 2.9% | Hepatomegaly | 7 | 35 | 20.0% |
| **Neurological syndrome** | | | | Conjunctival alterations | 7 | 36 | 19.4% |
| Meningeal irritation | 4 | 36 | 11.1% | | | | |
| Seizures | 2 | 36 | 5.6% | Conjunctival suffusion | 19 | 37 | 51.4% |

Care Unit (ICU). The presence of complications was 21.6% (8/37), with pulmonary complications being the most frequent (75.0%, 6/8). Among patients with successful follow-up after discharge, the fatality was 3.1% (1/32). This death was a 45-year-old female patient who had a DPO of 28 days at the time of enrollment and presented general edema and pulmonary complications before death.

Genotyping through 16S sequencing was attempted in all PCR-positive samples but it was successful in two. In both *Leptospira santarosai* was identified as the infecting species.

**Table 5. Laboratory findings in leptospirosis confirmed patients, Apartadó, 2016.**

| | n | % | mean | | n | % | mean |
|---|---|---|---|---|---|---|---|
| **Blood cell count** | | | | **Liver function** | | | |
| WBC > 11,000 mm3 | 7/33 | 21.2 | 8410 | ALAT > 40 U/L | 9/18 | 50.0 | 152 |
| Neutrophile > 60% | 20/33 | 60.6 | 63.2 | ASAT > 40 U/L | 9/19 | 47.3 | 84.1 |
| Neutrophiles> 70% | 16/33 | 48.4 | | Total bilirubin > 1.2 mg/dL | 5/13 | 38.4 | 1.1 |
| Hemoglobin < 11 g/dl | 7/33 | 21.2 | 12.1 | Total bilirubin/Direct bilirubin ratio | 6/12 | 91.6 | |
| Hematocrit < 33% | 9/33 | 27.2 | 36.3 | **Urine Chemistry** | | | |
| Platelets < 150,000 mm3 | 9/33 | 27.2 | 209.8 | Urine density < 1005 | 1/28 | 3.5 | 1014 |
| Platelets > 400,000 mm3 | 1/33 | 3.0 | | Urine pH < 5 | 0/28 | 0.0 | 5.9 |
| **Kidney function** | | | | Urine pH > 8 | 1/28 | 3.5 | |
| Creatinine > 1.1 mg/dL | 3/20 | 15.0 | 0.7 | **Inflammation** | | | |
| BUN > 24 mg/dL | 2/13 | 15.3 | 14.6 | CPK > 190 | 7/20 | 35.0 | 231.3 |

WBC: White blood cells, BUN: blood urea nitrogen

## 4. Discussion

The present study reports a frequency of leptospirosis confirmation using PCR of 37% among 100 febrile patients identified as probable for leptospirosis based on compatible cases and positive nondiagnostic serological tests. PCR is a suitable test for acute leptospirosis because if applied in the first week, it yields results quickly and can confirm the diagnosis without the need for a convalescent sample. This, in turn, leads to prompt treatment and the implementation of necessary public health interventions. PCR can detect patients early in the infection, before development of antibodies. PCR has also benefits over culture for diagnostic purposes due to sensitivity and speed as cultures can take between three to six months to yield results [23]. The frequency of confirmed diagnosis found among compatible clinical syndrome was slightly lower than the 58% documented by Pérez-García et al. for the Urabá region, however, that study used three different diagnostic methods including culture, MAT, and indirect immunofluorescence assay (IFA), combining direct and indirect tests [17].

The combination of a direct test like PCR, with an indirect test like MAT, has been proposed as a way to improve the diagnosis of leptospirosis [24]. Indirect or serological tests (MAT and IFA) allow for the detection of antibodies usually appearing 5–7 DPO and can diagnose patients that are no longer in the window for a positive blood PCR. However the sensitivity in a single sample is low, and the requirement paired samples diminishes their usefulness in a clinical context [25]. Optimal implementation of PCR for clinical diagnosis needs to consider illness duration and sample. Using blood samples, the study found 9/21 positive PCR results in week one of disease onset, 5/21 in week two, 2/21 in week three, and 8/21 beyond week three. Using urine samples, the study found 6/15 positive PCR results in week one of disease onset, 4/15 in week two, 0/15 in week three, and 5/15 beyond week three. Memory bias should be taken into account, as many patients can delay access to health services, and the possibility of overlapping or combined infections in tropical settings, these are not exclusive to the Urabá region, but, are common among rural and underdeveloped communities where the disease prevails [26–28]. These results put into question the time frame for PCR use, suggesting the detection window is longer than previously thought, at least, among patient that require hospital care.

With limitations on sensitive and specific diagnostic tests applicable in low complexity settings in a timely manner, it is important to improve the understanding of the clinical presentation of the disease, as many clinicians will have to rely solely on this to determine the need for antibiotic treatment and keeping patients within institutions for observation. The most frequently reported symptoms among the confirmed patients were headache (93.8%), chills and sweating (42.9%), nausea (75%), dizziness (74.3%), and vomiting (61.1%). Considering that a history of fever was required for inclusion, chills, and sweating are commonly associated with that symptom, and all may be regarded as highly non-specific. A study conducted in a tertiary institution in Cali, Colombia by Cedano et al. evidenced a high frequency of fever, reported by 85% of patients, and gastrointestinal symptoms, with nausea being reported by 67.8%, vomiting by 31%, stomachache by 52.9%, and diarrhea by 35.6%. However, the study was done on severe cases that required intensive care unit (ICU) admission [29]. In a retrospective study done by Jauréguiberry et al. in metropolitan France, where the incidence of leptospirosis is low, the most frequent symptoms reported among 34 cases identified from 1992 to 2002 were headache in 75%, myalgia in 55%, arthralgia in 45%, and vomiting in 39% of cases [30]. In a retrospective study done by Daher et al. in Brazil, with 201 patients recruited from 1985 to 2006 from a single metropolitan institution, the most frequently reported symptoms were myalgia in 92.5%, headache in 74.6%, vomiting in 71.6%, dehydration in 63,.5%, and chills in 62.2% of cases [31]. Leptospirosis presents itself as a highly unspecific febrile disease and

might be hard to differentiate from other acute febrile illnesses in a setting where they overlap. It should be considered as a differential diagnosis in patients with suspected arboviral infections, malaria, or other tropical diseases.

Analysis of factors associated with PCR positivity among all enrolled patients revealed that eye pain was negatively associated with a PCR-positive result. Retroorbital pain is a symptom frequently reported in febrile patients confirmed with dengue [32] and other arboviruses such as Zika [33], which are diseases endemic to the study area. Our findings suggest the presence of eye pain (retroorbital pain) could be considered as a symptom to guide the diagnosis away from leptospirosis (follow-up of PCR-negative patients did not include a final diagnosis). This does not apply to all ophthalmological symptoms, as other studies found that ophthalmological manifestations such as conjunctival congestion, chemosis, or subconjunctival hemorrhage, particularly in icteric febrile syndrome are highly pathognomonic of systemic leptospirosis in the acute stages of the infection [34,35].

Of all PCR-positive patients with reported creatinine results, there were few with signs of renal involvement (15.0% 3/20); however, the frequency of anuria/oliguria was higher (21.6%; 8/37). In the kidney, proximal tubules are the main lesion site, determined by morphological and immune-enzymatic changes of the brush border, among other abnormalities, leading to alterations in urine components before an alteration in glomerular filtration rate [36]. Echeverry-Toro et al. found renal function impairment in 54% of hospitalized patients of all age groups with leptospirosis in Medellin [37]. The differences in the organs most frequently affected during leptospirosis outbreaks might be related with the species involved. Despite our findings, evidence suggests that the kidney is still an important target during leptospirosis, and its involvement is probably one of the main factors leading to severe disease.

In the liver, *Leptospira* invades the intercellular junctions of host hepatocytes; subsequently, bile leaks from canaliculi and jaundice occurs [38]. Jaundice was reported in 8.3% of confirmed patients which is much lower than what has been reported in other studies, Biscornet et al. reported jaundice in 22% of leptospirosis patients, Cedano et al. in 27.6%, Jauréguiberry et al. in 34.4% and Daher et al. in 94.4% of patients [29–31,39]. The studies by Cedano et al., Jauréguiberry et al., and Daher et al. were conducted in metropolitan reference institutions of major cities, thus it would be expected for these institutions to treat more severe/complicated cases of leptospirosis (29–31). The study by Biscornet et al. was carried out with data from multiple governmental institutions, which might lead to a wider range of disease presentations, however as a retrospective study severe cases are more likely to have been studied, thus it is not strange than the percentage of jaundice was also higher than that found in the current study (39). Kidney and liver injury are indicators of severity and lead to icterohemorrhagic fever and renal impairment evidenced in Weil's Syndrome [40]. These organs should be monitored in patients with suspected leptospirosis, to detect complications and progression towards severe disease.

An important finding was that in children, inability to stand (PR 2.4; 95% CI: 1.09–5.26) and anuria/oliguria (PR 2.33; 95% CI: 1.22–4.45) were significantly associated with PCR positivity. The proportion of confirmed diagnoses was more than twice as high among children exhibiting these symptoms compared to those who did not. This is compatible with other findings that suggest that disease in children might favor kidney impairment over hepatic involvement [41]. Anacleto et al. reported a higher frequency of kidney injury due to leptospirosis in pediatric patients, with a significantly higher level of blood urea nitrogen (BUN) and serum creatine in oliguric subjects, finding that oliguric AKI due to leptospirosis is more frequent and severe than non-oliguric kidney failure in pediatric patients [42]. Kidney involvement among pediatric patients with undifferentiated febrile illnesses in tropical settings should lead to the inclusion of leptospirosis in the differential diagnosis.

Given the difficulties in obtaining opportune laboratory confirmation in a clinical setting, understanding how other laboratory tests change during leptospirosis is important in aiding clinicians towards its diagnosis. Among PCR-positive leptospirosis patients in the study, the test most frequently ordered tests by attending physicians was a blood cell count, and an elevated neutrophils count was present in 60.6%. The role of neutrophils in the response to bacterial infections is well established [43]. Leukocytosis and thrombocytopenia have been described as usual findings in leptospirosis; in the present study, platelet count was altered in 30.3% of patients with 27.3% having a value under 150,000. There is reported evidence of severe leptospirosis, with aplastic anemia and reports of spirochetes in bone marrow [44]. Approximately 50% of patients with acute leptospirosis usually have elevated enzyme creatine phosphokinase (CPK) during the first illness week [45]. Grau et al. concluded that CPK levels are of diagnostic aid, not only in the most severe diseases but also in the mild ones [46]. In the present study, an increased CPK was found in 35% of the confirmed cases.

Antibiotic selection by clinicians was generally adequate based on recommendations of the WHO for the treatment of severe and moderate cases with penicillin and doxycycline respectively. It should be noted that there are treatment alternatives for severe cases with third-generation cephalosporins, particularly ceftriaxone, or even cefotaxime as an alternative to penicillin G in critically ill patients [47].

Leptospirosis can lead to different complications, and clinicians need to understand them to diagnose and manage them. The proportion of patients with complications was 21.6% and the main complications were pulmonary. Lung involvement has been identified as a fatality indicator, but the occurrence tends to be low [48]. A recent study by Parra Barrera et al. reviewed lab-confirmed leptospirosis cases in Colombia between 2015 and 2020 using information from the epidemiological surveillance system, that study found that 82.2% of confirmed cases were severe cases, with pulmonary complications reported in 18.4% of reports [49]. The present study saw a relatively high number of hospitalizations, 83.8% this is in part explained by the fact that patients included in the study came from medical institutions with emergency departments that tend to treat more severe patients while redirecting mild cases to outpatient care facilities. Most cases of leptospirosis tend to be mild, however, studies tend to report high percentages of severe disease and hospitalization due to low diagnostic levels among mild cases [48]. Among patients with follow-up, lethality was 3.1%, this is higher than that recorded by Cedano et al. (2019), of 1.1% in severe cases treated in ICUs in the city of Cali, and lower than the 5% recorded by Echeverry-Toro et al. in the city of Medellin [29,37]. Parra Barrera et al. also reported Weil Disease in 18.4%, the absence of this syndrome among the present study's patients could be associated with changes in the virulence pattern or the specific pathology of circulating serovars that lead to other clinical presentations (49).

The genetic classification of species was not part of the original study protocol; however, it later became a possibility, due to sample size and quality genetic classification was only achieved in two samples, and in both cases, *Leptospira santarosai* was confirmed. Studies done in Taiwan, by Lin et al. and Wang et al., in which *Leptospira santarosai* was identified as the predominant species, reported a high incidence of pulmonary complications [35,50]. This suggests that *Leptospira santarosai* might be more pathogenic towards the lung. The study by Jauréguiberry et al. reported that 15% of cases, required admission into the ICU, 5.9% for AKI, and 2.9% for pulmonary complications [30]. In contrast in a study by Daher et al. pulmonary complications were reported in 14% of patients but dialysis was needed in 50.7% of patients and death occurred in 15.4% of cases [31]. *Leptospira santarosai* has been previously identified as a species involved in human leptospirosis in the region of Urabá, Colombia [51]. Animal studies using *Leptospira santarosai* species from Colombia showed it had the invasive capacity of *Leptospira interrogans* despite being associated with less severe clinical manifestations [52]. The

difference in the frequency of pulmonary vs kidney/hepatic complications in different regions might be influenced by the species of *Leptospira*.

### 4.1. Limitations

The current study is limited by sample size, and by the fact that treatment, laboratory work, and follow-up were left to the treating physicians and institutions. This led to incomplete complementary laboratory information. Future studies require enough funding to benefit from a cohort design, as patient flow in highly endemic areas is enough to justify it.

Selection bias was identified as most patients included lived in urban areas, given that rural inhabitants have geographical limitations in accessing health institutions. Initial clinical data was gathered upon patient inclusion in the early stages of care, reducing memory bias, later medical records were checked for additional information on patients' evolution.

## 5. Conclusion

The current study's findings match what has been previously described, with an acute septicemic phase of leptospirosis being characterized as a patient with fever, pain, and gastrointestinal symptoms. This unfortunately mimics many febrile illnesses such as dengue, malaria, chikungunya, Zika, acute toxoplasmosis, and hepatitis A, among others, common in regions with high leptospirosis endemicity. The "Classic case" of a two-phase febrile symptom that evolves into Weil's Disease was not frequent among the patients in this study, and even though, it adds to the knowledge surrounding the clinical presentation of leptospirosis, more studies like this are necessary to better characterize the disease.

## Supporting information

**S1 Data. Study data base.**
(XLSX)

## Author Contributions

**Conceptualization:** Claudia Munoz-Zanzi, Piedad Agudelo-Florez.

**Data curation:** Pablo Uribe-Restrepo, Janeth Perez-Garcia.

**Formal analysis:** Janeth Perez-Garcia, Claudia Munoz-Zanzi.

**Funding acquisition:** Janeth Perez-Garcia, Piedad Agudelo-Florez.

**Investigation:** Janeth Perez-Garcia, Margarita Arboleda, Piedad Agudelo-Florez.

**Methodology:** Pablo Uribe-Restrepo, Janeth Perez-Garcia, Margarita Arboleda, Claudia Munoz-Zanzi, Piedad Agudelo-Florez.

**Writing – original draft:** Pablo Uribe-Restrepo.

**Writing – review & editing:** Janeth Perez-Garcia, Margarita Arboleda, Claudia Munoz-Zanzi, Piedad Agudelo-Florez.

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
