## [Decision Letter · Decision Letter 0]

27 Jun 2024

Dear Dr Uribe-Restrepo,

Thank you very much for submitting your manuscript "Clinical presentation of human leptospirosis in febrile patients: Urabá, Colombia" for consideration at PLOS Neglected Tropical Diseases. As with all papers reviewed by the journal, your manuscript was reviewed by members of the editorial board and by several independent reviewers. The reviewers appreciated the attention to an important topic. Based on the reviews, we are likely to accept this manuscript for publication, providing that you modify the manuscript according to the review recommendations. 

Sincerely,

Yung-Fu Chang

Academic Editor

Elsio Wunder Jr

Section Editor

Reviewer's Responses to Questions

**Key Review Criteria Required for Acceptance?**

**Methods**

-Are the objectives of the study clearly articulated with a clear testable hypothesis stated?

-Is the study design appropriate to address the stated objectives?

-Is the population clearly described and appropriate for the hypothesis being tested?

-Is the sample size sufficient to ensure adequate power to address the hypothesis being tested?

-Were correct statistical analysis used to support conclusions?

-Are there concerns about ethical or regulatory requirements being met?

Reviewer #1: - The objective was clearly stated in the text. 

- The study design was well designed although a bit biased because patients were selected after positive serology was confirm at the clinic.

- The authors have addressed the weakness of a small sample size.

- Statistics and the analyzes was well done

Reviewer #2: The study is appropriately described/designed. While the sample size is small, it does not preclude publication. Analyses are appropriate. There are a couple of specific comments below pertaining to specific results. There are no ethical concerns.

**Results**

-Does the analysis presented match the analysis plan?

-Are the results clearly and completely presented?

-Are the figures (Tables, Images) of sufficient quality for clarity?

Reviewer #1: Yes, easier t9o follow. I like to bold the significant findings for readers to capture the main message but this may not be allowed by the journal. 

All Tables were easy to follow.

Reviewer #2: In Section 4:

Interesting why authors report adults >19 years rather than adults 18+years? I’m struggling to see the relevance of this age break. Is there an exposure-related or clinical reason to discuss age groups this way? It would be important to know how many were children. Recommend breaking the age groups down, even if just in Table 1, especially since later there is discussion about risk factors in children. (also at some point authors will need to specify if children are those <18 years or actually <20 are who they consider children in the analysis).

Sentence beginning "Three patients has positive urine…” is just a little tough to decipher at first read. Perhaps a rewording, something like … Of these patients, 22 were positive on blood sample, 12 were positive on urine, and three tested positive on both urine and blood...

In Section 4.1: 

Sentence beginning "Although it was not statistically significant .... " should be removed. If the analysis did not reach statistical significance, there is no evidence of the positivity being lower, or higher, than the referent group.

From Tables 2 and 3, I do not see fever or temperature recorded, but that would be important for the reader to assess – temperature, duration of fever? Also, maybe a reminder in this section that owed to inclusion criteria, all patients presented with fever.

Paragraph 2 references splenomegaly as associated with PCR positivity but this is not suggested by the table. Methods describe the test for significance as Fischer’s exact or chi-squared, and by the indicated Fischer’s exact p-value, this finding is not significant. (If the authors are instead alluding to a p-value derived from the PR regression, I would not place confidence in this with such a small group.) As I read the data, no signs or symptoms among these febrile patients were associated with testing positive for lepto by PCR. However, when stratified on age group, there were some features that stood out.

I suppose Table 3 should be referenced in text of paragraph 1.

A minor typo in the last sentence of 2nd paragraph reports PR as RP.

Section 4.2:

Can authors include temp, bpm, rpm in the descriptive tables so a comparison can be gleaned PCR pos vs neg, even if there were no differences? Similarly for the blood parameters? If these are available, these could prove useful in the clinical diagnosis when diagnostic testing is limited.

**Conclusions**

-Are the conclusions supported by the data presented?

-Are the limitations of analysis clearly described?

-Do the authors discuss how these data can be helpful to advance our understanding of the topic under study?

-Is public health relevance addressed?

Reviewer #1: Yes, conclusions addressed some of the strengths as well as limitations. Perhaps add on the benefits of these findings to advance clinical diagnosis of this disease in this endemic area.

Reviewer #2: Discussion:

First sentence, recommend inserting the word “100 febrile patients” – it is not necessary but is a helpful reminder to readers that the cohort is febrile.

I believe authors could highlight that they demonstrate the utility of PCR testing as a confirmatory test in lepto, including over a longer timeframe in the clinical course. 

Sentence beginning “Indirect or serological tests…” is long, and I suggest for clarity breaking the sentence at “.. positive blood PCR. However, the sensitivity (of PCR? Serology?) .. is low … requirement of paired samples for serologic diagnosis limits usefulness in clinical context.” As the sentence reads it is a little hard to follow which tests the author is talking about for each reference.

Second paragraph, it seems in the results that positivity was highest in the latest-DPO group. Perhaps instead of listing how many samples tested positive, refer to these as the proportion positive, a more useful glimpse into what the authors are describing. PCR may be a useful diagnostic tool over a longer period of time during clinical illness than is traditionally employed. Why? Could this be because patients presenting later are more severe cases with higher pathogen load?

Paragraph 3, first sentence, recommend change sensible to sensitive “With limitations on sensitive and specific …”

Highlighting the main presenting symptoms is useful, but in this study, these symptoms did not distinguish lepto confirmed PCR-positive cases from the probable cases.

Paragraph 4, I think this is the first mention of eye pain in the paper referencing a statistically significant association. But as the paper reads, no signs or symptoms were associated with lepto PCR positivity. Perhaps I am missing something in the results section, but I am also not finding this in the tables… If this is a finding of the study, authors should present the data – it could be valuable. 

Paragraph 5, can the authors suppose why their finding on kidney function is not consistent with other reports? Could this relate to delayed clinical presentation, after AKI resolved? Whatever the hypothesis, this would be a good discussion point.

Paragraph 7, is it meant to say the interesting finding is inability to stand also, since the paragraph focuses on kidney injury. It is agreed that the importance of kidney injury in leptospirosis cannot be overstated.

Paragraph 8, sentence about “increase in neutrophils”, just to clarify, was there a longitudinal assessment or is this referring to a high number of neutrophils measured at enrollment? Not to be nit-picking, but just to confirm that the reader is not confused. Is it the authors hypothesis that CPK could help distinguish lepto cases from some other tropical diseases?

Paragraph 10, It would be interesting to comment on the 84% hospitalization in the context outside of Colombia – it sems a high proportion of severe cases, given that lepto is considered generally a mild disease. Perhaps authors can comment on this in the context of what is known about lepto severity, in general, or at least in the greater region. Is this an artifact of who presents for care (more severe cases?) or is it suggested that there truly is a high prevalence of severity among lepto infections? I think some discussion on this point is warranted.

Conclusion:

I suggest to authors that they consider highlighting also that the “classic” signs were rare and perhaps no longer tell-tale of lepto, and also that while signs & symptoms were not useful overall, there may be some utility among specific age groups, such as clinical features pointing to lepto in children, different from those features in adults. Do the authors think it is also important to draw conclusion on disease severity?

As a general comment for the discussion, it would be nice for the authors to place the results in greater context, including outside of Colombia or the region.

**Editorial and Data Presentation Modifications?**

Reviewer #1: - Reference 20 - missing year of publication

Reviewer #2: The authors present % to the tenth decimal place. With such a small population size (n=37 for the subgroup most analyzed), it may be appropriate to round to the whole number since there is such little confidence in the decimal digit.

Similarly, the PR while an appropriate estimate, are difficult to interpret with such a small population size. This reporting is not necessary to change, but too much emphasis should not be placed on these confidence intervals. Authors may wish to acknowledge that the ability to draw estimates is limited and external validity is not yet understood. There are some areas where the text perhaps could be reduced, more concisely presented. I do not think this is a reason to reject the article, but the authors may take another pass at some of the results and discussion.

**Summary and General Comments**

Reviewer #1: Strengths:

- The area of study in Colombia which is known to be endemic for many other febrile illness of unknown nature

- The team is well known in the area and have built several studies on the epidemiology of leptospirosis in this area.

- The characterization of diagnostic signs/symptoms for leptospirosis in this endemic area

Weaknesses:

- The sample size

- Serology is still used in remote health clinics and PCR is not widely used. What there any concordance between serology and PCR positive results?

- Are there potential recommendations for clinicians based on the findings of signs/symptoms?

- The potential date of patient's visit may not correlate with the onset of signs/symptoms or the day of exposure. This information is self-reported and can become a confusing.

Reviewer #2: This is overall a nicely written article of clinical and public health importance. I wondered if the authors considered a continuation of their research to provide a greater sample size - this comment is not meant to point out a weakness of the present study. I appreciate the opportunity to review the article and have provided I believe a thorough review that I hope helps the authors to consider appropriate minor edits. Lepto continues to be a concern, including in endemic areas, yet receives little attention or updates to clinical understanding. The authors provide valuable information to the scientific community.

PLOS authors have the option to publish the peer review history of their article (what does this mean?). If published, this will include your full peer review and any attached files.

Reviewer #1: Yes: Fernando P Monroy

Reviewer #2: No

Figure Files:

Data Requirements:

Reproducibility:

References

---

## [Decision Letter · Decision Letter 1]

11 Aug 2024

Dear Dr Uribe-Restrepo,

We are pleased to inform you that your manuscript 'Clinical presentation of human leptospirosis in febrile patients: Urabá, Colombia' has been provisionally accepted for publication in PLOS Neglected Tropical Diseases.

Best regards,

Yung-Fu Chang

Academic Editor

Elsio Wunder Jr

Section Editor

**Key Review Criteria Required for Acceptance?**

**Methods**

-Are the objectives of the study clearly articulated with a clear testable hypothesis stated?

-Is the study design appropriate to address the stated objectives?

-Is the population clearly described and appropriate for the hypothesis being tested?

-Is the sample size sufficient to ensure adequate power to address the hypothesis being tested?

-Were correct statistical analysis used to support conclusions?

-Are there concerns about ethical or regulatory requirements being met?

Reviewer #2: (No Response)

**Results**

-Does the analysis presented match the analysis plan?

-Are the results clearly and completely presented?

-Are the figures (Tables, Images) of sufficient quality for clarity?

Reviewer #2: (No Response)

**Conclusions**

-Are the conclusions supported by the data presented?

-Are the limitations of analysis clearly described?

-Do the authors discuss how these data can be helpful to advance our understanding of the topic under study?

-Is public health relevance addressed?

Reviewer #2: (No Response)

**Editorial and Data Presentation Modifications?**

Reviewer #2: (No Response)

**Summary and General Comments**

Reviewer #2: These authors responded appropriately to the reviewers' comments and present a nice, well-written revised manuscript. Recommend to accept the article without further revisions.

PLOS authors have the option to publish the peer review history of their article (https://journals.plos.org/plosntds/s/editorial-and-peer-review-process#loc-peer-review-historywhat does this mean?). If published, this will include your full peer review and any attached files.

**Do you want your identity to be public for this peer review?** For information about this choice, including consent withdrawal, please see our https://www.plos.org/privacy-policyPrivacy Policy.

Reviewer #2: **Yes: **Rebecca SB Fischer

---

## [Editor Report · Acceptance letter]

13 Sep 2024

Dear Dr Uribe-Restrepo,

We are delighted to inform you that your manuscript, "Clinical presentation of human leptospirosis in febrile patients: Urabá, Colombia," has been formally accepted for publication in PLOS Neglected Tropical Diseases.

Best regards,

Shaden Kamhawi

co-Editor-in-Chief

Paul Brindley

co-Editor-in-Chief
